# Structural and Magnetic Properties of Inverse-Heusler Mn_2_FeSi Alloy Powder Prepared by Ball Milling

**DOI:** 10.3390/ma15030697

**Published:** 2022-01-18

**Authors:** Ondřej Životský, Kateřina Skotnicová, Tomáš Čegan, Jan Juřica, Lucie Gembalová, František Zažímal, Ivo Szurman

**Affiliations:** 1Department of Physics, Faculty of Electrical Engineering and Computer Science, VŠB—Technical University of Ostrava, 17. listopadu 2172/15, 708 00 Ostrava, Czech Republic; lucie.gembalova@vsb.cz; 2Regional Materials Science and Technology Centre, Faculty of Materials Science and Technology, VŠB—Technical University of Ostrava, 17. listopadu 2172/15, 708 00 Ostrava, Czech Republic; katerina.skotnicova@vsb.cz (K.S.); tomas.cegan@vsb.cz (T.Č.); jan.jurica@vsb.cz (J.J.); ivo.szurman@vsb.cz (I.S.); 3Institute of Physics of Materials, Czech Academy of Sciences, Zizkova 22, 616 00 Brno, Czech Republic; zazimal@ipm.cz; 4Department of Physical Electronics, Faculty of Science, Masaryk University, Kotlarska 267/2, 611 37 Brno, Czech Republic

**Keywords:** inverse-Heusler alloys, ball milling, powders, microstructure, Néel temperature, magnetic transitions

## Abstract

Ternary Mn_2_FeSi alloy was synthesized from pure elemental powders by mechanical alloying, using a high-energy planetary ball mill. The formation of an inverse-Heusler phase after 168 h of milling and subsequent annealing at 1173 K for 1.5 h was confirmed by X-ray diffraction. The diffractogram analysis yielded XA structure and the lattice parameter 0.5677 nm in a good agreement with the theoretically obtained value of 0.560 nm. The final powder was formed by particles of irregular shape and median diameter D50 of 3.8 μm and their agglomerates. The chemical analysis resulted in the mean composition of 49.0 at.% Mn, 25.6 at.% Fe and 25.4 at.% Si. At room temperature, the prepared samples featured a heterogeneous magnetic structure consisting of dominant paramagnetic phase confirmed by Mössbauer spectrometry and a weak ferro-/ferrimagnetic contribution detected by magnetization curves. From the field-cooled and zero-field-cooled curves the Néel temperature of 67 K was determined.

## 1. Introduction

The current rapid development of technology increases the requirements for materials research and the search for new material compositions. One of the attractive and currently investigated classes are Heusler alloys, which contain more than 4000 potential members and are applied in the fields like spintronics [1], magnetooptics [2,3], magnetoresistance and magnetocaloric devices [4,5], topological materials [6], shape memory alloys [7,8], spin injectors [9], etc. In general, regular types of Heusler materials are ternary intermetallic compounds with composition of elements X_2_YZ (full-Heusler) or XYZ (half-Heusler), where X and Y are transition metals (e.g., Fe, Co, Mn) and Z is an s, p element, e.g., Si, Al, In, Sn. The full-Heusler alloy consists of two binary compounds, XY and XZ, both of CsCl type crystal structure ordered into *L*2_1_ or *B*2 structures. If two of four fcc sublattices are occupied by X atoms (Figure 1a), the X_2_YZ alloy is of *L*2_1_ ordering in space group $Fm\bar{3}m$ (225). The Wyckoff coordinates of X elements in *L*2_1_ structure are (0, 0, 0) and (½, ½, ½), Y elements (¼, ¼, ¼) and Z elements (¾, ¾, ¾). If one fcc sublattice remains unoccupied the general formula XYZ represents half- or semi-Heusler alloy being of *C*1*_b_* structure. An exchange of X and Y elements in the formula of the full-Heusler alloy results in an inverse-Heusler alloy (Figure 1b) characterized by XA structure, space group $F\bar{4}$*3m* (216).

Currently, two basic directions in the research of Heusler alloys can be observed. The first one is focused on the way how to synthesize these materials. In addition, the conventional techniques including arc and induction melting [10,11] producing the bulk materials, or dc- and rf-sputtering on different types of substrates in a form of thin films [12], there are still efforts for searching Heusler alloys with improved properties leading to involvement of non-traditional preparation technologies. Therefore, in last few years, there is couple of papers showing interesting properties of Heusler alloys prepared either by mechanical alloying (MA) in a form of powders [13,14] or by planar flow casting (PFC) technique having ribbon shapes [15,16]. The various ways of synthesis lead to structural imperfections on the atomic scale (antisite atoms, vacancies, etc.), different structure morphologies, different chemically ordered phases and subsequently to different physical properties determining the various applications of Heusler alloys as mentioned briefly above.

The second direction is the production of Heusler alloys with new compositions. Currently, there are many papers showing theoretical ab initio or first-principles calculations and suggesting new Heusler alloy compositions with interesting magnetic and electronic properties [17,18]. So-called inverse-Heusler alloys form one of such groups. They are characterized by a formula X_2_YZ and own the XA phase in space group $F\bar{4}$*3m* (216) (Figure 1b) that differs from *L*2_1_ in the occupation of B and C sites. Even in this case the papers with theoretical calculations prevail [19,20] and if the alloys are produced, then mainly by mentioned conventional technologies [21,22]. 

The present studies are focused on the inverse-Heusler Mn_2_FeSi alloys. Theoretical calculations performed on this system [19,23] determined a lattice constant of 0.560 nm, total magnetic moment of about 2 µ_B_/f.u. (ferrimagnetic order, half-metallicity) and relatively large and antiparallel Mn moments at sites A and B. Experimentally, this alloy has so far been prepared only in the form of ingots and thin films [21,24]. However, observed antiferromagnetic behavior at low temperatures contradicts theoretical calculations and refers to the need for further detailed research [25]. This paper is focused on the Mn_2_FeSi alloy prepared by mechanical alloying into a powder form and on its detail structural and physical characterization. For this purpose, the X-ray diffraction, electron microscopy, Mössbauer spectrometry and room- and low-temperature magnetic measurements are used and the results are compared with experimental and theoretical literature data.

## 2. Materials and Methods

### 2.1. Materials

The powders of manganese (≥99.3 wt.%), iron (≥99.5 wt.%) and silicon (99.9985 wt.%) from Alfa Aesar (Massachusetts, USA) were used as raw materials. The data on particle size distribution of these powders are given in Table 1. The Mn:Fe:Si ratio was 50:25:25 (in at.%).

### 2.2. Synthesis

The inverse-Heusler Mn_2_FeSi alloy was synthetized by mechanical alloying using ball milling followed by thermal treatment. Mn, Fe and Si powders were initially mixed using the Turbula T2F mixer (Willy A. Bachofen AG, Muttenz, Switzerland) rotated at 50 rpm for 15 min. The powder mixture with initial weight of 50 g (56.7 wt.% Mn, 28.8 wt.% Fe, 14.5 wt.% Si) was placed in a stainless-steel milling bowl together with stainless-steel balls of 5 mm in diameter. The ball-to-powder weight ratio was 10:1. The n-hexane was added as the solvent in order to reduce the surface energy of particles and achieve a small particle size as well as good homogeneity. The milling bowl was sealed, evacuated several times and then filled with argon gas (99.999%). The process was realized using a high-energy ball mill E-max (Retsch GmbH, Haan, Germany) at 500 rpm for 168 h. Controlled integrated water cooling of milling bowl was used during milling. The alloying process was checked after selected times of milling. The pure elements were still observed after 80 h of milling as it is shown in Figure 2. Nevertheless, after 168 h of milling no pure elements were detected but the X-ray diffraction (XRD) measurement did not show any typical pattern expected for Mn_2_FeSi alloy. On the contrary, only a few sharper peaks next to broad amorphous halo were seen. This was caused by a presence of n-hexane used during milling. Therefore, a small amount of powder was thermally treated (1173 K/1.5 h) to remove solvent which was confirmed by repeated XRD measurement. The same annealing conditions, 1173 K/1.5 h and the dynamic argon atmosphere with flow rate 2 L/min, in a high-temperature sintering furnace (Xerion Advanced Heating^®^ Ofentechnik GmbH, Berlin, Germany) were subsequently used to form final (annealed) Mn_2_FeSi alloy.

### 2.3. X-ray Diffraction Analysis

The crystal structure and phase composition of powders were studied by powder X-ray diffraction. The XRD diffractometer AXS D8 Advance (Bruker AXS GmbH, Karlsruhe, Germany) equipped with the position-sensitive LynxEye detector and CuKα radiation at room temperature (RT) was used for measurements in the range 2*θ* = 20°–95°, steps 0.014° and time per step 2 s. The qualitative and quantitative analyses of diffractograms were performed using the Rietveld method.

### 2.4. Microstructural Characterization and Particle Size Distribution

The microstructure, chemical composition and element mapping were studied by scanning electron microscopy (SEM) using a QUANTA 450 FEG (FEI Company, Hillsboro, OR, USA) equipped with an energy-dispersive X-ray (EDX) APOLLO X analyser. The images were taken in the backscattered electron mode (BSE). The particle size distribution was measured by laser diffraction particle size analyser MasterSizer 3000 (Malvern Panalytical Ltd., Malvern, UK).

### 2.5. Mössbauer and Magnetic Measurements

Mössbauer spectrometry (MS) measurements were performed in transmission geometry by γ-rays at RT using a ^57^Co (Rh) source. Calibration of the velocity scale was done with α-Fe at RT and the isomer shifts are given with respect to its Mössbauer spectrum. All spectra were evaluated using the transmission integral approach in the program CONFIT [26]. The experimental points were analysed by the double- and/or single-line components yielding values of isomer shift (δ) and quadrupole splitting (Δ). The relative representations of these subcomponents are denoted by A.

Magnetic properties were obtained using a vibrating sample magnetometer (VSM) EZ9 (Microsense, Lowell, MA, USA) and a Physical Property Measurement System (PPMS, Model P935A, San Diego, CA, USA) Quantum Design. VSM was used for measurements of room- and high-temperature magnetization curves with maximal applied magnetic field of ±1600 kA/m (±2 T). PPMS was applied for low-temperature experiments, these are (i) magnetization curves at chosen constant temperatures with maximal magnetic field of ±2400 kA/m (±3 T) and (ii) field-cooled (FC) and zero-field-cooled (ZFC) curves measured in the temperature range 2–293 K at a magnetic field of 80 kA/m.

## 3. Results and Discussion

### 3.1. Morphology, Phase and Chemical Composition

The particle morphologies of initial manganese, iron and silicon powders are shown in Figure 3. Mn powder consisted of irregular polyhedral particles with sharp edges. Fe powder is characterized by a nearly spherical shape of particles with smooth surface. Si powder was typical by the irregular shaped particles with rough surface and high porosity. The annealed Mn_2_FeSi alloy powder (Figure 4) was characterized by the irregular shape of particles and bimodal distribution with peaks at 2.9 μm and 13.6 μm, respectively. Its median particle diameter D50 reached a value of 3.8 μm.

The chemical composition of the final Mn_2_FeSi powder, determined from the volume of particles (i.e., metallographic cut of powder sample was prepared), is summarized in Table 2 as averaged values from seven-spot EDX analysis. It can be seen that the chemical composition corresponds very well to the nominal composition of the alloy. The distribution of Mn, Fe and Si, investigated using the X-ray element mapping, is uniform throughout the particle volume and no obvious inhomogeneities were observed (see Figure 5).

X-ray diffraction patterns in Figure 6 show visible difference between the as-milled (a) and annealed (b) states of samples. The diffractogram of as-milled powder (Figure 6a) contains broad halos between 30° and 40° and similarly at higher 2*θ* angles, that next to only few sharp peaks come from the organic n-hexane used during milling. Consequently, these halos diminish after the sample annealing and the Rietveld analysis of its diffractogram (Figure 6b) evidences cubic Heusler alloy with the lattice parameter 0.5677 nm well agreed with the theoretical value 0.560 nm of Mn_2_FeSi alloy [18,22]. Generally, it is not simple to distinguish the inverse- and full-Heusler structure in their diffractograms. Nevertheless, the theoretical simulations yield higher relative intensity of the (111) diffraction peak compared to (200) for the inverse-Heusler alloy of XA structure [21]. Based on this result, the pattern in Figure 6b confirms formation of the inverse-Heusler Mn_2_FeSi alloy. Next to peaks characterizing the produced Heusler alloy, the peaks at positions 35°, 41° and 43° reflect; moreover, a presence of not specified impurity phase formed during the alloying process. It could be iron and/or manganese oxides speculatively identified from Mössbauer spectrum analysis based on Ref. [27] and presented below. 

### 3.2. Magnetic and Mössbauer Results

Figure 7 shows the room temperature Mössbauer spectrum (a) and magnetization curves of the annealed Mn_2_FeSi alloy powder measured at room and elevated temperatures (b). The experimental points (x) of the Mössbauer spectrum were analysed using single-line (singlet) and two-line (doublet) subcomponents of different intensities and different hyperfine parameters summarized in Table 3. The main contribution of the singlet (53.6%) represents a paramagnetic Mn_2_FeSi phase while doublets can be speculatively ascribed to a nonstoichiometric (Fe,Mn)-oxides and/or to chemical disorder at agglomerate boundaries. No presence of ferromagnetic component was identified by this Mössbauer measurement.

RT magnetization curve (Figure 7b, 303 K) is not saturated and shows linear dependence of magnetization on an applied external magnetic field *H* for *H* > ±150 kA/m. This confirms paramagnetic nature of the alloy powder being in full agreement with Mössbauer analysis. However, magnetization reversal in the low external magnetic field, *H* < ±150 kA/m, characterized by nonzero remanent magnetization, *M_r_* ≈ 0.015 Am^2^/kg, and coercivity, *H_c_* ≈ 4 kA/m, reflects a presence of a small contribution of ferro- and/or ferrimagnetic (distorted) phase. Heterogeneous magnetic behavior is observed also on magnetization curves at elevated temperatures, whereas magnitudes of *M_r_* and *H_c_* parameters decrease with increasing temperature. 

The low-temperature magnetic measurements of inverse-Heusler Mn_2_FeSi alloy powder are presented in Figure 8. The FC and ZFC curves shown in Figure 8a detect low magnetization of alloys when cooled from room temperature to about 170 K. The magnetic behavior is similar to that at room temperature and the magnetization curves in this temperature range (Figure 8b, 200 K) consist of a weak magnetization reversal at low magnetic fields and linear contribution at higher fields. 

The FC and ZFC curves separate at approximately 155 K. This temperature corresponds to the inflection point of both curves and can be denoted as the thermodynamic temperature of a transition to ferro-/ferrimagnetic state. We suppose that splitting of the FC/ZFC curves is not caused by the distorted phase, but by the impurities that exhibit dynamic behavior similar to the superparamagnetic systems. As a consequence, a more pronounced increase of remanent magnetization and coercive field is seen (Figure 8c). Therefore, the superposition of two magnetic contributions below 155 K, namely, ferro-/ferrimagnetic phase and disordered phase, is expected. Detailed analyses of the magnetization and FC-ZFC curves reveal an increase of disordered phase contribution during cooling below 155 K. 

The decrease of disordered phase occurs only after exceeding the temperature of 67 K, which corresponds to the Néel temperature, *T_N_*, below which the alloy exhibits an antiferromagnetic behavior. Contrary, the ferro-/ferrimagnetic phase represented by the coercive field and remanent magnetization continue to increase even during cooling to 2 K. The highest value of magnetization at a temperature of 2 K and a magnetic field of 2400 kA/m is about 12 Am^2^/kg. Generally, the theoretically predicted magnetic structure giving the lowest total energy for the XA crystallographic structure at zero temperature is ferrimagnetic. However, experimentally confirmed antiferromagnetic behavior can be realized in tetragonal or cubic (frustrated) crystallographic structures. 

Magnetic properties of prepared Mn_2_FeSi alloy powder are different from annealed ingots (773 K/48 h) and powders (1223 K/100 h) presented in the literature [21,28]. Both systems exhibit antiferromagnetic-paramagnetic transition at *T_N_* about 50 K. However, it can be seen from Ref. [28] that an increase in the amount of Fe at the expense of Mn can significantly change the magnetic behavior of the alloy powder. For example, the system Mn_1.5_Fe_1.5_Si [28] shows antiferromagnetic-ferromagnetic-paramagnetic transitions with Néel temperature at 62 K and Curie temperature at 137 K, but the shape of the FC-ZFC curves in the ferromagnetic region is completely different from that in Figure 8a, where we suppose superposition of ferro-/ferrimagnetic and paramagnetic contributions (67–155 K). Thus, it seems that the final magnetic properties will depend not only on the composition of the alloy, but also on the temperature and annealing time, which is significantly shorter (1.5 h) in the currently prepared alloy powders. 

## 4. Conclusions

The present work is devoted to the microstructural and magnetic investigations of Mn_2_FeSi alloy prepared from pure elemental powders by mechanical alloying. The formation of the inverse-Heusler (XA) phase with the corresponding lattice parameter (0.5677 nm) required ball milling for 168 h in the presence of solvent (n-hexane) and subsequent annealing in a sintering furnace at 1173 K for 1.5 h in a protective argon atmosphere. SEM/EDX and particle size distribution analyses of the annealed powder indicated practically no deviations from the nominal composition, irregular shape of particles having bimodal distribution with peaks at 2.9 μm and 13.6 μm and median particle diameter D50 of 3.8 μm. 

Magnetic measurements revealed heterogeneous magnetic behavior of the final alloy. In the temperature range from 170 K to 573 K we detected the dominant paramagnetic (disordered) phase and a small ferro-/ferrimagnetic contribution showing magnetization reversal and nonzero values of remanent magnetization *M_r_* and coercive field *H_c_* on measured magnetization curves. Below 155 K, the step increase of *M_r_* and *H_c_* and the splitting of the FC and ZFC curves are observed. A decrease in the disordered phase contribution is found after the system has cooled below 67 K. This is reflected in the maximum of the FC and ZFC curves at 67 K that probably corresponds to the Néel temperature below which the antiferromagnetic arrangement of the spins originates. 

The present investigations have evidenced a possibility to prepare the inverse-Heusler Mn_2_FeSi alloy using mechanical alloying. Nevertheless, several open questions as, e.g., an origin of the ferro-/ferrimagnetic phase, its effect on the magnetic behavior of the pure inverse-Heusler Mn_2_FeSi, the reason why it was not detected by highly sensitive Mössbauer spectrometry, the real influence of producing technology, etc., remain for the future investigations. 

## Figures and Tables

**Figure 1 materials-15-00697-f001:**
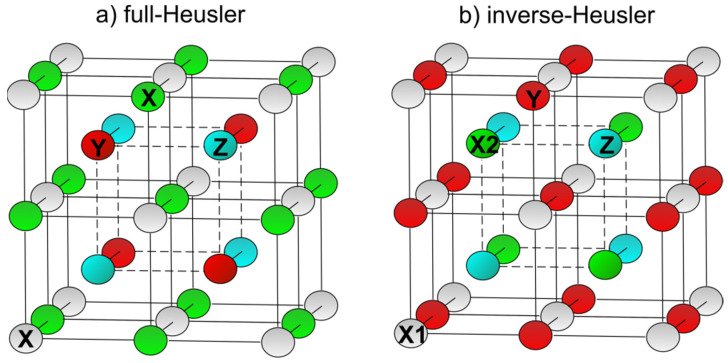
Unit cells formed by A, B, C and D fcc sublattices of (**a**) regular full-Heusler (*L*2_1_) X_2_YZ and (**b**) inverse-Heusler (XA) X_2_YZ alloys. In the full-Heusler alloy X = X_A_ = X_C_ (Y_B_ and Z_D_), whereas in the inverse-Heusler alloy X1 = X_A_ and X2 = X_B_ (Y_C_ and Z_D_).

**Figure 2 materials-15-00697-f002:**
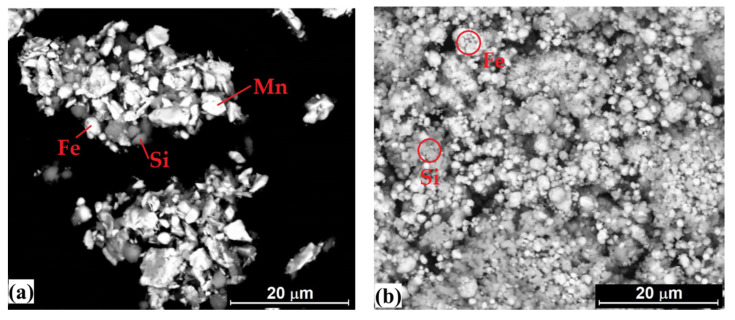
The particle morphology of Mn-Fe-Si powder after 1 h (**a**) and 80 h (**b**) of milling.

**Figure 3 materials-15-00697-f003:**
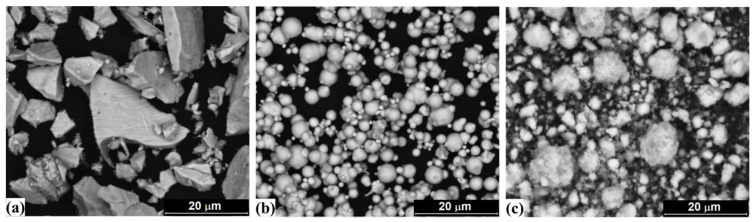
The particle morphologies of initial powders: (**a**) Mn, (**b**) Fe, (**c**) Si (SEM/BSE).

**Figure 4 materials-15-00697-f004:**
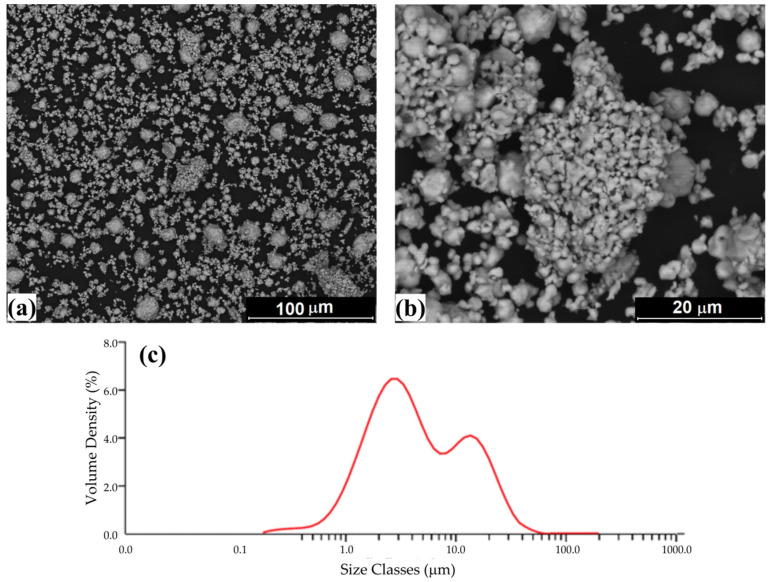
(**a**,**b**) The particle morphology of inverse-Heusler Mn_2_FeSi alloy powder (SEM/BSE) and (**c**) its particle size distribution with bimodal character.

**Figure 5 materials-15-00697-f005:**
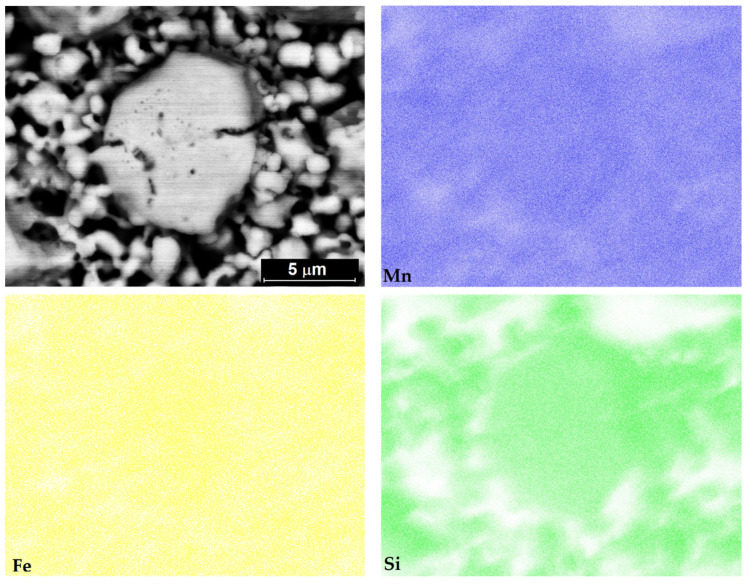
X-ray maps showing the distribution of Mn, Fe and Si in the particle of annealed Mn_2_FeSi alloy powder.

**Figure 6 materials-15-00697-f006:**
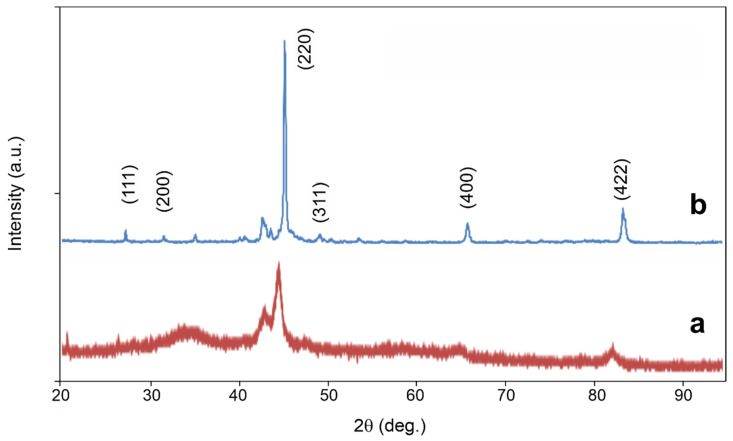
X-ray diffraction pattern of Mn_2_FeSi alloy powder milled for 168 h (**a**) and annealed at 1173 K for 1.5 h (**b**).

**Figure 7 materials-15-00697-f007:**
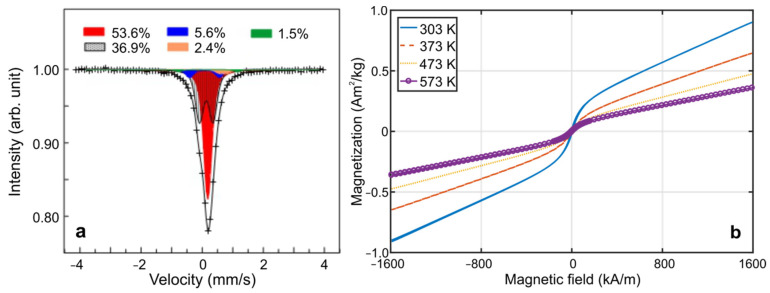
(**a**) Room temperature Mössbauer spectrum. (**b**) Magnetization curves of the annealed Mn_2_FeSi powder at room and elevated temperatures.

**Figure 8 materials-15-00697-f008:**
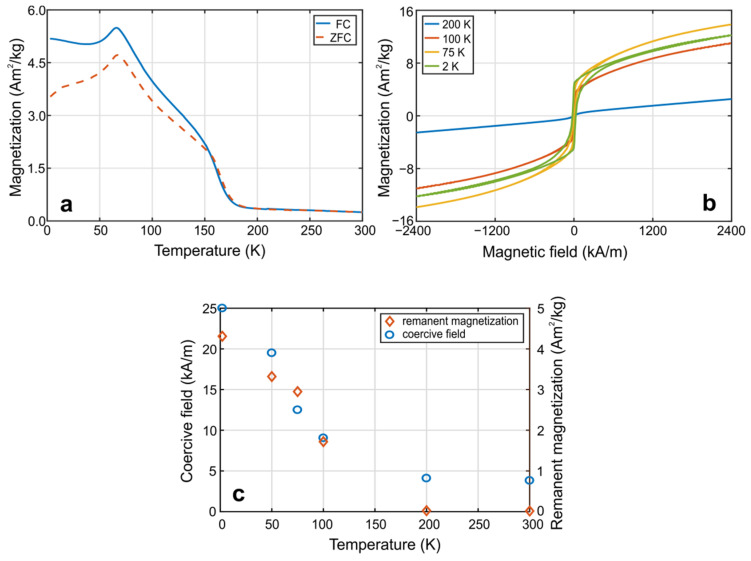
Low-temperature magnetic properties of annealed Mn_2_FeSi powder sample. (**a**) Field-cooled (FC) and zero-field-cooled (ZFC) curves at a constant magnetic field of 80 kA/m, (**b**) magnetization curves at different temperatures, (**c**) coercive field and remanent magnetization as a function of temperature.

**Table 1 materials-15-00697-t001:** Particle size distributions of starting Mn, Fe and Si powders.

Element	D10 (μm)	D50 (μm)	D90 (μm)
**Mn**	5.6	29.8	67.6
**Fe**	2.9	6.4	14.0
**Si**	1.0	5.5	16.8

**Table 2 materials-15-00697-t002:** The average chemical composition of inverse-Heusler Mn_2_FeSi alloy powder.

Mn (at.%)	Fe (at.%)	Si (at.%)
49.0 ± 0.8	25.6 ± 1.0	25.4 ± 1.5

**Table 3 materials-15-00697-t003:** Hyperfine parameters, isomer shift, δ, quadrupole splitting, Δ and relative intensity, A, of annealed Mn_2_FeSi powder.

Component	δ (mm/s)	Δ (mm/s)	A (%)
S (singlet)	0.181(1)		53.6(5)
D1 (doublet)	0.133(3)	0.445(7)	36.9(4)
D2	0.078(16)	0.974(33)	5.6(3)
D3	0.696(26)	0.261(33)	2.4(5)
D4	0.196(26)	1.922(56)	1.5(2)

## Data Availability

The data presented in this study are available on request from the corresponding author.

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
