# Peer review of "Structural and Magnetic Properties of Inverse-Heusler Mn2FeSi Alloy Powder Prepared by Ball Milling"

_materials, 2022, doi:10.3390/ma15030697_

Round 1

Reviewer 1 Report

The authors present characterization data on powdered Mn2FeSi that was produced by a combination of ball milling and high temperature annealing.  The paper is well-written and interesting.  I have a couple of questions and one formatting suggestion.

First, I suggest that the authors expand on lines 104/105 where they refer to, 'a series of own experiments'.  What data/results from the own experiments were used to decide that 168 hours was the correct milling time and that 1173K/1.5 hours were the appropriate corresponding annealing conditions?  For example, reference [22] (which the authors use for comparison of magnetic properties) explains that a specific set of data was presented due to the clarity of its x-ray data.

Second, do the authors have any hypotheses about the origin of the differences in behaviors between the mechanically-milled and arc-melted samples (commented on in lines 251-254)?  A direct comparison is beyond the scope of the paper.  However, hypotheses about potential origins of the differences based on the authors' own investigations and experiences with the material could be illuminating for readers.

Finally, I realize that both Figure 7b and Figure 8b are intended to have the same label on the y-axis as 7a and 8a, respectively.  However, for clarity, I suggest the label be restated so that each figure is complete on its own.

Author Response

We thank the reviewer for his/her useful comments. The text below provides answers to the reviewer questions and comments. The changes are highlighted in red in the revised version of the manuscript.

Comment

First, I suggest that the authors expand on lines 104/105 where they refer to, 'a series of own experiments'.  What data/results from the own experiments were used to decide that 168 hours was the correct milling time and that 1173K/1.5 hours were the appropriate corresponding annealing conditions?  For example, reference [22] (which the authors use for comparison of magnetic properties) explains that a specific set of data was presented due to the clarity of its x-ray data.

Answer

We have agreed with this suggestion. The process of Mn2FeSi alloy production is described in more details in the subsect. 2.2 and the X-ray pattern of as-milled alloy after 168 h is included in Fig. 6.

Comment

Second, do the authors have any hypotheses about the origin of the differences in behaviors between the mechanically-milled and arc-melted samples (commented on in lines 251-254)?  A direct comparison is beyond the scope of the paper.  However, hypotheses about potential origins of the differences based on the authors' own investigations and experiences with the material could be illuminating for readers.

Answer

The results presented in the paper clearly show that the main difference between the Mn2FeSi ingots and powders lies in the existence of the ferro-/ferrimagnetic phase. Its existence is probably connected with the used preparation technology, postpreparation treatment (temperature and time of annealing), different microstructure, etc. We have prepared also our own ingots, where the ferro-/ferrimagnetic phase is not seen at room temperature, but originates after splitting the FC-ZFC curves at lower temperatures (results will be soon published). This would suggest that the increase in this phase after splitting of the FC-ZFC curves comes from the Mn2FeSi system and not from the parasitic ferro-/ferrimagnetic phase. It is evident that further research is needed and many things have yet to be clarified. In addition, we are also working on theoretical calculations and plan to publish all new results gradually.  

Comment

Finally, I realize that both Figure 7b and Figure 8b are intended to have the same label on the y-axis as 7a and 8a, respectively.  However, for clarity, I suggest the label be restated so that each figure is complete on its own.

Answer

Both figures were corrected according to the reviewer comment.

Reviewer 2 Report

The manuscript “Structural and magnetic properties of inverse-Heusler…” by O. Životský et. al. deals with production, structural and magnetic characterization of Mn2FeSi powder from powders of elemental components. An initial step of the fabrication is long-term ball milling. The study seems to be motivated by previously reported discrepancy between the magnetic ordering of the conventionally fabricated alloy (arc melting) and theoretically predicted ground state. The main concern of the reviewer is apparently inhomogeneous structure of the fabricated powder. Despite rather clear evidence of this heterogeneity seen by XRD, Mössbauer and magnetic properties, the authors either try to ignore/minimize its role or attribute a number of crucial observations to the properties of the material as such. The problem then becomes to match the observations of, say, magnetic hysteresis or low-field nonlinearity with assumed para- or antiferromagnetic states. As a result, a number of important observations are ignored and many interpretations lack clarity. Numerical estimations made are doubtful. The reviewer believes that without sorting out all additional ordering transitions and identifying their origin, the manuscript cannot be accepted. Therefore, it requires major revision.

General comments

What is the origin of the abrupt increase of magnetization below ca. 400 K?

Do the particles of a few micron in size contain grains, line 200??? How is this confirmed?

What does a “small degree of ordering” that accounts for the low-field hysteresis mean, line 214? It is ascribed to Mn2FeSi system, line 216. It cannot be paramagnetic, but slightly ferro- or ferrimagnetic. Most likely, this is the consequence of the heterogeneous structure of the powder, a part is paramagnetic, a part has magnetic order.

Application of the Curie-Weiss law to treat data that show hysteresis is not justified. The Curie temperature determined is not trustworthy.

“Gradual bifurcation”, line 237, is nonsense.

The interpretation of the emerging difference between FC and ZFC curves as due to “blocked state of the largest particles”, line 238, does not sound and clarifies nothing.  

Lines 239, 240: The alloy (which one – Mn2FeSi?) becomes “more ordered”, but still paramagnetic! This is the same contradiction as in interpretation of low-field hysteresis.

Dominant <220> peak does not indicate inverse structure, line 188.

FC and ZFC curves merge in the vicinity of clear shoulder of M(T). If the AFM contribution is subtracted, this shoulder might yield a maximum, typical for freezing into the glassy state.

Low temperature hysteresis, Fig. 8b, has very little to do with the AFM behavior.

In general, magnetic properties point to the heterogeneous structure of the produced alloy powder (confirmed also by XRD and Mössbauer), a part of the particles do undergo AFM ordering, others – ferri – or ferromagnetic one. The experimental data are superposition of probably AFM and ferrimagnetic patterns. The energies of AFM and ferrimagnetic structures are nearly degenerated.

The reviewer doubts thus that the production of a heterogeneous structure that does not uniform magnetic properties is a real success as declare the authors. The fabrication method must be modified. As cast alloys and films show better properties.

Specific comments

  • First sentence of the abstract, line 19, is wrong. (i) Ball milling itself does not produce Heusler alloy, the Heusler structure is formed by annealing. (ii) What do the authors mean by “success”, what is it´s measure? The fabricated powder is not homogeneous, the homogeneity of the fabricated powder is much worse than of the bulk cast material.
  • The meaning of “powder alloy” is obscure, it is definitely not the same as “alloy powder”. What is fabricated actually is “alloy powder”.
  • 10 line 76 is not in a right place
  • Line 77, “for the first time” should be skipped
  • The first sentence, line 95, should be skipped. “Powder metallurgy” says nothing and is not needed in this section.
  • Figure 7c should become figure 7a; 7a and 7b - 7b and 7c.
  • 27, line 191, is a PhD thesis. The reference to the entire thesis is not adequate.
  • The tendency in the magnetic hysteresis below TN are discussed for the data at 50 and 2 K, line 249, 250. However, the data at 50 K are not shown. This is not a good style.

Author Response

We thank the reviewer for his/her useful comments. The text below provides answers to the reviewer questions and comments. The changes are highlighted in red in the revised version of the manuscript.

Comment

What does a “small degree of ordering” that accounts for the low-field hysteresis mean, line 214? It is ascribed to Mn2FeSi system, line 216. It cannot be paramagnetic, but slightly ferro- or ferrimagnetic. Most likely, this is the consequence of the heterogeneous structure of the powder, a part is paramagnetic, a part has magnetic order.

Answer

We agree with the reviewer. The description of the magnetic properties of powder at room temperature in the second paragraph of section 3.2 should have such a meaning. We would prefer the statement that the powder has heterogeneous structure, a part is paramagnetic, a part has magnetic order. Origin of the ferro-/ferrimagnetic contribution is not clear and can come from oxide phases detected by XRD, boundaries of agglomerates, etc. However, it should be emphasized that the ferro-/ferromagnetic contribution is weak. This is evident from the FC-ZFC curves in the temperature range from 170 K to 303 K, which coincide and have the shape typical of a paramagnetic material. Also, the Mössbauer spectrum does not detect any sextet typical of the presence of a Fe-containing ferromagnetic phase. Based on these facts, the second paragraph of section 3.2 was rewritten.

Comments

Application of the Curie-Weiss law to treat data that show hysteresis is not justified. The Curie temperature determined is not trustworthy.

What is the origin of the abrupt increase of magnetization below ca. 400 K?

Answer

The temperature scan in Figure 7b was shown to see where and in what temperature range the Curie temperature was determined. We agree with the reviewer that even in the temperature range above 400 K, a weak contribution of the ferro-/ferrimagnetic phase is observed on the hysteresis loops. Due to this contribution, the determination of the Curie temperature using the Curie-Weiss law is burdened with a certain error. Therefore, both the temperature scan (in the old version of the paper Figure 7b) and the corresponding part of the Curie temperature text have been removed.

Till now we do not have any explanation for abrupt increase of magnetization below 400 K.

Comment

Do the particles of a few micron in size contain grains, line 200??? How is this confirmed?

Answer

SEM measurements did not reveal the presence of grains, but rather agglomerations of particles. The text was corrected.

Comment

“Gradual bifurcation”, line 237, is nonsense.

Answer

This phrase has been deleted.

Comments

The interpretation of the emerging difference between FC and ZFC curves as due to “blocked state of the largest particles”, line 238, does not sound and clarifies nothing.

Lines 239, 240: The alloy (which one – Mn2FeSi?) becomes “more ordered”, but still paramagnetic! This is the same contradiction as in interpretation of low-field hysteresis.

FC and ZFC curves merge in the vicinity of clear shoulder of M(T). If the AFM contribution is subtracted, this shoulder might yield a maximum, typical for freezing into the glassy state.

Low temperature hysteresis, Fig. 8b, has very little to do with the AFM behavior.

Answer

Last paragraph of section 3.2 devoted to the low-temperature magnetic properties was rewritten.

 Comment

Dominant <220> peak does not indicate inverse structure, line 188.

Answer

We agree with the reviewer. It is difficult to distinguish between the inverse-Heusler (XA) and full-Heusler (L21) phase in the XRD diffractogram. For the inverse-Heusler structure, the relative intensity of (111) peak should be higher than that of (200) peak. This is fulfilled in our case. This information is newly added into the last paragraph of section 3.1.

Comment

In general, magnetic properties point to the heterogeneous structure of the produced alloy powder (confirmed also by XRD and Mössbauer), a part of the particles do undergo AFM ordering, others – ferri – or ferromagnetic one. The experimental data are superposition of probably AFM and ferrimagnetic patterns. The energies of AFM and ferrimagnetic structures are nearly degenerated.

Answer

We agree with the reviewer; the heterogeneous magnetic properties of the prepared alloy powder are clearly seen. The main conclusions are the following. In the temperature range 2 - 67 K it seems that we have some non-collinear magnetic ordering (local ferromagnetic phase like spirals), but macroscopically – at longer distances – it behaves as AFM (magnetic moment is compensated). However, these preliminary results from theoretical calculations must be proved by other experimental results. Heating the sample above 67 K leads to a decrease in both the ferro-/ferrimagnetic contribution and the disordered phase (we have precisely analysed FC-ZFC curves and measured magnetization curves above 67 K). It should also be noted that the FC-ZFC curves show a superposition of both contributions and at a relatively small magnetic field of 80 kA/m, the ferro-/ferrimagnetic phase plays an important role in the temperature range from 67 K to 155 K (relatively high values of Hc and Mr). Above the 155 K FC-ZFC curves converge, the change in slope is probably due to a step drop in the ferro-/ferrimagnetic contribution, while the disordered phase indicates a dominant paramagnetic behavior.

Comment

First sentence of the abstract, line 19, is wrong. (i) Ball milling itself does not produce Heusler alloy, the Heusler structure is formed by annealing. (ii) What do the authors mean by “success”, what is it´s measure? The fabricated powder is not homogeneous, the homogeneity of the fabricated powder is much worse than of the bulk cast material.

Answer

We agree with the reviewer; the first sentence of the Abstract was corrected. Also word “successfully” was removed from the Abstract, Introduction, and Conclusion. Nevertheless, this is the first experimentally prepared Mn2FeSi alloy powder with a dominant Heusler phase. The results presented in the paper clearly show that the main difference between the Mn2FeSi ingots and powders lies in the existence of the ferro-/ferrimagnetic phase. Its existence is probably connected with the used preparation technology, postpreparation treatment (temperature and time of annealing), different microstructure, etc. We have prepared also our own ingots, where the ferro-/ferrimagnetic phase is not seen at room temperature, but originates after splitting the FC-ZFC curves at lower temperatures (results will be soon published). This would suggest that the increase in this phase after splitting of the FC-ZFC curves comes from the Mn2FeSi system and not from the parasitic (residual) ferro-/ferrimagnetic phase. Of course, we will also try to remove the ferro-/ferrimagnetic phase by changing the annealing time and temperature. It is evident that further research is needed and many things have yet to be clarified. In addition, we are also working on theoretical calculations and plan to publish all new results gradually.  

Last thing, the reviewer compares the homogeneity of the ingot samples from Reference [21] (A. Aryal et al. Jalcom 823 2020 153770) prepared by arc melting with the alloy powders from this paper. The FC-ZFC curves show clearly paramagnetic behaviour of both ingots and alloy powders at room temperature. The presence of a weak ferro-/ferrimagnetic phase in alloy powders is seen only from hysteresis loops at room and elevated temperatures. However, the authors of reference [21] do not show any such loops in their work. So is the ferro-/ferrimagnetic phase present in their samples at room temperature or not?

Comment

The meaning of “powder alloy” is obscure, it is definitely not the same as “alloy powder”. What is fabricated actually is “alloy powder”.

Answer

This was corrected in the whole paper.

Comment

10 line 76 is not in a right place

Answer

Citation number was changed to [25].

Comment

Line 77, “for the first time” should be skipped

Answer

It was corrected.

Comment

The first sentence, line 95, should be skipped. “Powder metallurgy” says nothing and is not needed in this section.

Answer

The first sentence of section 2.2 was modified.

Comment

Figure 7c should become figure 7a; 7a and 7b - 7b and 7c.

Answer

The subplots of Figure 7 were interchanged.

Comment

27, line 191, is a PhD thesis. The reference to the entire thesis is not adequate.

Answer

We have added the year of the thesis and the pages corresponding to the description of Mn2FeSi alloys to the citation [27].

Comment

The tendency in the magnetic hysteresis below TN are discussed for the data at 50 and 2 K, line 249, 250. However, the data at 50 K are not shown. This is not a good style.

Answer

The magnetization curve at 50 K was of course measured, corresponding magnetic parameters (coercive field and remanent magnetization) were and are presented in Fig. 8c. We do not show this curve in Fig. 8b, because it practically overlaps with the magnetization curve at 75 K. Figure 8b would thus become confusing for the readers. In addition, the section devoted to the magnetic properties at low temperatures has been completely reworked.

Round 2

Reviewer 2 Report

The revised manuscript is less confusing than the initial version. Still, at least two points need clarification.

  • The interpretation of the “blocking temperature” at ca. 155 K remains unclear (lines 235-239). How is the size of the particles involved? Obviously, this effect relates with the magnetically ordered part of the material. There is no way the effect can be ascribed to the paramagnetic state. Then, the contribution of the magnetically ordered material is not that small, as is postulated, e.g. , in the abstract.  

The emerging difference of FC and ZFC patterns is a canonical effect in ferroics. The authors should clearly state whether or not they attribute this effect to para- or ferri/ferromagnetic states. In the former case, a viable explanation should be given.

  • Before, the reviewer suggested considering co-existence of practically degenerated ferri- and antiferro- ordering in the fabricated material. The authors still try to reduce their observations to the antiferromagnetic state, as has been reported for a cast alloy. The authors should be more independent here, especially taking into account that ferrimagnetic  is the theoretical ground state. Then the authors do not need to look for hypothetical impurities, oxides, etc.  For example, variations of composition that measure a few percent (Table 2) can stimulate variation of magnetic ordering in the same “parent” material.

Below are minor technical and linguistic comments

Lines 20-22 should read:

The formation of an inverse-Heusler phase after 168 hours of milling and subsequent annealing at 1173 K for 1.5 hours was confirmed by X-ray diffraction.

Line 28: “…magnetization curves”

Line 67: “ab initio” in italics

Line 110: what is the meaning of “…the same annealing conditions...”?

Line 205: “at agglomerate boundaries”

The sentence on lines 103-104 should be re-phrased, 500 rpm and 168 hrs do not refer to cooling of the milling bowl.

Author Response

We thank the reviewer for his/her useful comments and also for English corrections. Changes in the paper are highlighted in red. The main changes were made in the section devoted to low-temperature magnetic measurements.

Comment

The interpretation of the “blocking temperature” at ca. 155 K remains unclear (lines 235-239). How is the size of the particles involved? Obviously, this effect relates with the magnetically ordered part of the material. There is no way the effect can be ascribed to the paramagnetic state. Then, the contribution of the magnetically ordered material is not that small, as is postulated, e.g., in the abstract.  

The emerging difference of FC and ZFC patterns is a canonical effect in ferroics. The authors should clearly state whether or not they attribute this effect to para- or ferri/ferromagnetic states. In the former case, a viable explanation should be given

Before, the reviewer suggested considering co-existence of practically degenerated ferri- and antiferro- ordering in the fabricated material. The authors still try to reduce their observations to the antiferromagnetic state, as has been reported for a cast alloy. The authors should be more independent here, especially taking into account that ferrimagnetic  is the theoretical ground state. Then the authors do not need to look for hypothetical impurities, oxides, etc.  For example, variations of composition that measure a few percent (Table 2) can stimulate variation of magnetic ordering in the same “parent” material.

Answer

After many discussions, we decided to present the magnetic results as follows. The temperature of 155 K, where the FC-ZFC curves separate, corresponds to the inflection point of both curves and can be denoted as the thermodynamic temperature of a transition to ferro-/ferrimagnetic state. We suppose that splitting of the FC/ZFC curves is not caused by the distorted phase (ferro-/ferrimagnetic phase detected on the magnetization curves at room temperature and at 200 K), but by the impurities that exhibit dynamic behavior similar to the superparamagnetic systems. Nevertheless, we still think that below 155 K you can see the superposition of the mentioned ferro-/ferrimagnetic contribution from impurities and the paramagnetic contribution coming from the Mn2FeSi alloy. The first reason is that the Mn2FeSi systems presented in the literature show always antiferromagnetic-paramagnetic behavior with a Néel temperature of about 50 K (we added a new reference number [28]). Moreover, in Ref. [28] we can find system Mn1.5Fe1.5Si that shows antiferromagnetic -ferromagnetic-paramagnetic transitions with Néel temperature at 62 K and Curie temperature at 137 K, but the shape of the FC-ZFC curves in the ferromagnetic region is completely different from that in Fig. 8a, where we suppose superposition of ferro-/ferrimagnetic and paramagnetic contributions (67-155 K). Thus, it seems that the final magnetic properties will depend not only on the composition of the alloy, but also on the temperature and annealing time, which is significantly shorter (1.5 h) in currently prepared alloy powders. So the next step will be the preparation of a series of Mn2FeSi powders depending on the increasing annealing time and their detailed characterization.

We have also made progress in theoretical calculations. The theoretically predicted magnetic structure giving the lowest total energy for the XA crystallographic structure at zero temperature is ferrimagnetic [19,23]. However, experimentally confirmed antiferromagnetic behavior can be realized in tetragonal [21] or in cubic (frustrated) [our new unpublished results] crystallographic structures. Calculations are still ongoing.

Comment

Below are minor technical and linguistic comments

 Lines 20-22 should read:

The formation of an inverse-Heusler phase after 168 hours of milling and subsequent annealing at 1173 K for 1.5 hours was confirmed by X-ray diffraction.

Line 28: “…magnetization curves”

Line 67: “ab initio” in italics

Line 110: what is the meaning of “…the same annealing conditions...”?

Line 205: “at agglomerate boundaries”

The sentence on lines 103-104 should be re-phrased, 500 rpm and 168 hrs do not refer to cooling of the milling bowl.

Answer 

We accepted all comments and corrected the relevant sentences and phrases.
